# Altered Expression of Intestinal Tight Junctions in Patients with Chronic Kidney Disease: A Pathogenetic Mechanism of Intestinal Hyperpermeability

**DOI:** 10.3390/biomedicines12020368

**Published:** 2024-02-05

**Authors:** Georgia-Andriana Georgopoulou, Marios Papasotiriou, Pinelopi Bosgana, Anne-Lise de Lastic, Eleni-Evangelia Koufou, Evangelos Papachristou, Dimitrios S. Goumenos, Periklis Davlouros, Eleni Kourea, Vasiliki Zolota, Konstantinos Thomopoulos, Athanasia Mouzaki, Stelios F. Assimakopoulos

**Affiliations:** 1Division of Nephrology, Department of Internal Medicine, Medical School, University of Patras, 26504 Patras, Greece; georgop91@gmail.com (G.-A.G.); mpapasotiriou@yahoo.com (M.P.); epapadoct@hotmail.com (E.P.); dgoumenos@upatras.gr (D.S.G.); 2Department of Pathology, Medical School, University of Patras, 26504 Patras, Greece; bosgana.p@gmail.com (P.B.); hkourea@yahoo.com (E.K.); zol@med.upatras.gr (V.Z.); 3Laboratory of Immunohematology, Division of Hematology, Department of Internal Medicine, Medical School, University of Patras, 26504 Patras, Greece; delastic@gmail.com (A.-L.d.L.); mouzaki@upatras.gr (A.M.); 4Division of Cardiology, Department of Internal Medicine, Medical School, University of Patras, 26504 Patras, Greece; elenikoufou17@gmail.com (E.-E.K.); pdav@med.upatras.gr (P.D.); 5Division of Gastroenterology, Department of Internal Medicine, Medical School, University of Patras, 26504 Patras, Greece; kxthomo@hotmail.com; 6Division of Infectious Diseases, Department of Internal Medicine, Medical School, University of Patras, 26504 Patras, Greece

**Keywords:** chronic kidney disease, endotoxin, intestinal barrier, occludin, tight junctions

## Abstract

Background: Systemic inflammation in chronic kidney disease (CKD) is associated (as a cause or effect) with intestinal barrier dysfunction and increased gut permeability, with mechanisms not yet fully understood. This study investigated different parameters of the intestinal barrier in CKD patients, especially tight junction (TJ) proteins and their possible association with systemic endotoxemia and inflammation. Methods: Thirty-three patients with stage I–IV CKD (n = 17) or end-stage kidney disease (ESKD) (n = 16) and 11 healthy controls underwent duodenal biopsy. Samples were examined histologically, the presence of CD3+ T-lymphocytes and the expression of occludin and claudin-1 in the intestinal epithelium was evaluated by means of immunohistochemistry, circulating endotoxin concentrations were determined by means of ELISA and the concentrations of the cytokines IL-1β, IL-6, IL-8, IL-10 and TNF-α in serum were measured using flow cytometry. Results: Patients with stage I–IV CKD or ESKD had significantly higher serum endotoxin, IL-6, IL-8 and IL-10 levels compared to controls. Intestinal occludin and claudin-1 were significantly decreased, and their expression was inversely correlated with systemic endotoxemia. Regarding occludin, a specific expression pattern was observed, with a gradually increasing loss of its expression from the crypt to the tip of the villi. Conclusion: The expression of occludin and claudin-1 in enterocytes is significantly reduced in patients with CKD, contributing to systemic endotoxemia and inflammatory responses in these patients.

## 1. Introduction

Systemic inflammation without evident clinical infection is common in CKD patients [1,2,3]. This can exacerbate various pre-existing comorbidities such as cardiovascular disease, malnutrition and erythropoietin-resistant anemia, leading to increased morbidity and mortality [4,5,6]. Systemic inflammation is manifested in CKD patients by increased levels of pro-inflammatory cytokines and biomarkers of oxidative stress in the blood [7]. The translocation of bacteria and endotoxins from the gut due to injury to the uremic gut barrier plays a key role in promoting systemic inflammation [8,9,10,11,12]. On the other hand, proinflammatory cytokines exert injurious effects on the gut barrier’s integrity, further promoting endotoxin translocation and systemic inflammation, thus creating a vicious cycle [1,5,8,9,10,11]. Previous studies have shown that the intestinal barrier is structurally and functionally impaired in CKD, leading to increased gut permeability [11,13].

The main regulator of paracellular permeability is the apical junctional complex, which consists of TJs, the subjacent adherens junctions and desmosomes [14,15,16]. It serves as the main natural barrier against the influx of harmful contents in the internal environment [14,17]. To date, four groups of proteins have been identified as integral components of the TJs; occludin, members of the claudin family, junctional adhesion molecules (JAMs), and tricellulin. These proteins are associated with the actin cytoskeleton via an intracellular “tight junctional plaque”, which is mainly formed by members of the zonula occludens (ZO) protein family [18,19]. Occludin and claudins play a central role in the function of epithelial TJs, which dynamically regulate the passage of ions, macromolecules and cells through the paracellular pathway [20,21]. Previous studies using intestinal epithelial cell cultures and experimental animal models have shown that enterocyte TJ proteins occludin and claudin-1 are structurally and functionally disrupted after exposure to uremic conditions [22,23,24].

The aim of this study was to investigate whether the expression of the key TJ proteins occludin and claudin-1 is altered in the intestinal mucosa of patients with stage I–IV CKD or ESKD, contributing to systemic endotoxemia and inflammatory responses.

## 2. Materials and Methods

### 2.1. Study Design

This is a single-center, two-arm, non-randomized prospective study. Patients were eligible for inclusion if they were ≥18 years old and had been diagnosed with CKD. Exclusion criteria included pregnancy, severe cardiac valve insufficiency, heart failure (with a left ventricular ejection fraction of less than 50% or diastolic dysfunction), malignancies, chronic gastrointestinal diseases (e.g., Helicobacter pylori gastro-duodenitis, celiac disease, irritable bowel syndrome, inflammatory bowel disease) or bowel surgery, and the presence of any of the following conditions in the last four weeks: infections, alcohol abuse, gastrointestinal bleeding, pancreatitis, and treatment with drugs that could interfere with TJ regulation such as antibiotics, corticosteroids, non-steroidal anti-inflammatory drugs and antioxidants (vitamins C and E, allopurinol and N-acetyl cysteine).

All samples were collected and analyzed within a 24-month period. In total, 44 of the 60 patients examined met the above criteria. The study group consisted of patients with CKD documented by a previous renal biopsy or an eGFR (CKD-EPI calculation) below 90 mL/min/1.73 m^2^ or albuminuria of more than 30 mg in 24 h urine (n = 33). This group was divided into two groups: patients with stage I–IV CKD (CKD group, n = 17) and patients with ESKD receiving maintenance hemodialysis (HD) or peritoneal dialysis (PD) (ESKD group, n = 16). The CKD stage was classified using the CKD-EPI formula. The control group consisted of subjects without CKD who did not fulfill any of the above exclusion criteria (control group, n = 11).

All subjects enrolled in this study underwent an upper gastrointestinal (GI) endoscopy under light sedation, due to symptoms of dyspepsia, after consultation with a gastroenterologist, without any pathological findings. During endoscopy, three duodenal biopsies were taken from the second part of the duodenum. Only samples with a negative Helicobacter pylori biopsy were considered for participation in this study.

The study was approved by the Ethics Committee of the University Hospital of Patras, Greece (approval number: 480/4216) and was performed in accordance with the Declaration of Helsinki as revised in 2013. Written informed consent was obtained from all subjects prior to participation in the study.

### 2.2. Endotoxin and Cytokines Measurements

Before the endoscopy, a blood sample was taken for endotoxin and cytokine measurement. In the ESKD group, blood samples were obtained before a mid-week HD session or after drainage and before filling the peritoneal cavity with peritoneal dialysate in subjects on PD. The samples were collected in endotoxin-free vials and the serum was separated and stored at −80 °C until processing. The endotoxin concentration was measured by ELISA using the Human Endotoxin (ET) kit (cat#abx051541; Abbexa Ltd., Cambridge Science Park, Cambridge, UK) (test range 0.015–1.0 EU/mL, sensitivity < 0.005 EU/mL). The samples were processed after appropriate dilution according to the manufacturer’s instructions.

Measurement of the concentration of cytokines IL-1β, IL-6, IL-8, IL-10 and TNF-α in serum samples was performed with a BD FACS Array Bioanalyzer using a cytometric bead array (CBA) assay (Human Inflammatory Cytokines Kit, cat#551811, BD Biosciences, San Diego, CA, USA) (test range 20–5000 pg/mL, sensitivity 7.2 pg/mL for IL1β, 2.5 pg/mL for IL6, 3.6 pg/mL for IL8, 3.3 pg/mL for IL10 and 3.7 pg/mL for TNF-α).

### 2.3. Histopathological Evaluation and Immunohistochemistry

All biopsies were formalin-fixed and paraffin-embedded, and they were placed in formalin within 10 min from the resection and processed within 48 h. In each hematoxylin and eosin (H&E)-stained slide, several histologic features were evaluated in a blind fashion and recorded. These features included architectural distortion, villous blunting, surface and crypt epithelial injury, the presence and cell type of inflammation of the lamina propria, surface and cryptal intraepithelial infiltration, lamina propria fibrosis and granulation tissue formation. Apoptotic bodies are defined as round vacuoles containing fragments of karyorrhectic nuclear debris distinct from small, isolated fragments of nuclear chromatin and intraepithelial neutrophils or lymphocytes. Apoptotic bodies were counted in all architecturally well oriented consecutive crypts of the sample, regardless of crypt orientation, and their number per 100 intestinal epithelial cells is referred to as the apoptotic body count. For villus length (mm), at least 10 well-oriented villi were evaluated in each sample.

For immunohistochemistry (IHC), serial 3 μm tissue sections were cut, fixed in poly-L lysine-coated slides and further processed. The sections were initially dried at 25 °C for 24 h, deparaffinized in xylene and hydrated in gradient alcohol. Antigens were retrieved in Tris/EDTA buffer (pH 9) for 12 min with a pressure antigen retrieval procedure. Endogenous peroxidase activity was then blocked by incubating the sections in an endogenous peroxidase blocking solution (0.3% H_2_O_2_) at room temperature for 10 min. The sections were then incubated with the following primary antibodies: claudin-1 (rabbit polyclonal antibody, 1:100, WA314099, cat# 51–9000 Invitrogen, Rockford, IL, USA), occludin (rabbit polyclonal antibody, 1:80, VL314100, cat#71–1500, Invitrogen, Rockford, IL, USA), CD20 (monoclonal antibody, Clone L26, 1:200, Dako Carpinteria, Carpinteria, CA, USA) and CD3 (rabbit polyclonal antibody, 1:300, Dako, Glostrup, Denmark). Dako EnVision polymer (Dako EnVision Mini Flex, Dako Omnis, Angilent Technology Inc., Santa Clara, CA, USA, K8023) was used for signal detection. Diaminobenzidine (Dako Omnis, GV823, Glostrup, Denmark) was used as a chromogen and Harris hematoxylin was used for nuclear counterstaining. Positive and negative controls for antibody validation were used according to the manufacturer’s instructions.

Occludin, claudin-1 and CD3 immunohistochemical expression was recorded as present (+) or absent (−). For occludin expression, ten well-oriented villi were randomly selected per case and a percentage value of occludin (+) enterocytes was obtained by dividing the number of cells stained positive by the total number of enterocytes lining villi. For each part of the villi (crypt, middle and tip), a percentage value of occludin (+) enterocytes was also obtained by dividing the number of cells stained positive by the total number of enterocytes present in each part. The localization of the stain (nuclear, membranous, cytoplasmic) was also assessed. Moreover, the number of CD3(+) intraepithelial lymphocytes per 100 intestinal epithelial cells was recorded in ten well-oriented villi per case. Regarding claudin-1, at least 20 well-preserved crypts were evaluated in each case and a percentage value of claudin-1(+) enterocytes was obtained by dividing the number of cells stained positive by the total number of cryptal enterocytes. Photomicrographs were taken with cellSens Entry by Olympus on an Olympus BX41 microscope (Olympus Europa SE & Co., Wendenstraße 20, 20097 Hamburg, Germany). Two expert pathologists (PB and VZ) blinded to the pathological and clinical characteristics of all cases performed the histopathological and immunohistochemical analyses, and when scores between the two observers were discordant, a consensus was achieved by conference at a two-headed microscope.

### 2.4. Statistical Analysis

Data were analyzed using the SPSS statistical package for Windows (version 25.0; IBM, Armonk, NY, USA) and GraphPad Prism (version 9.1.0, GraphPad Software Inc., San Diego, CA, USA). Normality of data was tested using the Shapiro–Wilk test. Comparisons were performed using nonparametric analysis of variance (Kruskal–Wallis test) followed by a post hoc Mann–Whitney U test with Bonferroni correction (non-normally distributed data) or with one-way analysis of variance followed by a post hoc Tukey test (normally distributed data). The results are expressed as the median (interquartile range) for non-normally distributed data or the mean ± standard deviation (SD) for normally distributed data. The chi-squared test, with Yates’ correction if required, was used to compare the proportional data. Correlations were estimated by a nonparametric Spearman correlation test. All tests were two-tailed and a *p*-value of less than 0.05 was considered significant.

## 3. Results

### 3.1. Patients’ Characteristics

Patients’ clinical and biochemical characteristics are summarized in Table 1.

There were no significant differences concerning basic clinical characteristics of patients among groups except for serum creatinine, urea and eGFR.

### 3.2. Endotoxin Concentrations

Patients with either CKD or ESKD showed significantly higher endotoxin serum levels in comparison to controls (*p* < 0.01, Figure 1).

There was no significant difference in serum endotoxin levels between patients with CKD and ESKD.

### 3.3. Cytokine Levels

Cytokines levels in all groups are presented in Table 2.

Serum IL-1β and TNF-a levels did not differ significantly between groups. Serum IL-6 was significantly increased in both groups, CKD and ESKD, as compared to the controls (*p* < 0.01, respectively), and the same applied for IL-8 (*p* < 0.05, *p* < 0.001, respectively) and IL-10 (*p* < 0.001, *p* < 0.01, respectively). There were no significant differences between stage I–IV CKD and ESKD groups for all measured cytokines.

### 3.4. Intestinal Histopathology

Overall, the duodenal architecture appeared normal, and epithelial continuity was retained in all patients with CKD. Subepithelial edema was occasionally observed at the tips of some villi in patients with ESKD. The duodenal crypts were well preserved in all patients. No statistically significant differences were found between CKD or ESKD patients and the control group in apoptotic body count and villus length (Table 3).

### 3.5. Immunohistochemistry for TJ Proteins and Intraepithelial CD3(+) T-Lymphocytes

In healthy controls, occludin was expressed as membranous immunostaining, mainly at the apical part of epithelial cells, whilst granular cytoplasmic and subnuclear distribution was also observed. In the control group, almost all epithelial cells lining the villi and the epithelial cells of the crypts showed positive immunostaining for occludin (Figure 2A).

In patients with stage I–IV CKD (Figure 2B) and ESKD (Figure 2C), the expression of occludin was greatly reduced in numerous epithelial cells lining the villi (*p* < 0.001 as compared to controls, respectively) (Table 4, Figure 3).

Interestingly, in the CKD and ESKD groups, we observed a gradient of occludin immunostaining positivity along the length of the villi, from crypt to tip; occludin expression was maintained in the crypts and basal portion of the villi, and was reduced in the middle part of the villi in both groups, CKD and ESKD (*p* < 0.001, *p* < 0.01, respectively), while greater loss of its expression was observed at the tip (*p* < 0.001, respectively) (Table 4).

In healthy subjects, claudin-1 expression was observed in Brunner glands and crypts and it was absent in intestinal villi. Staining in Brunner glands did not show any significant difference between the CKD, ESKD and control groups. However, claudin-1 expression in crypts was significantly decreased in the CKD and ESKD groups compared to the controls (*p* < 0.001 and *p* < 0.01, respectively) (Figure 4).

There was no significant difference in intraepithelial CD3(+) T-lymphocytes amongst groups (Table 2).

### 3.6. Correlations

In all CKD patients (CKD stage I–IV and ESKD), the expression of occludin and claudin-1 in the intestinal mucosa (immunohistochemical semi-quantification) was significantly inversely correlated with endotoxin concentration (r = −0.616, *p* < 0.001 for occludin and r = −0.417, *p* < 0.01 for claudin-1).

## 4. Discussion

Intestinal mucosa injury, bacterial overgrowth and gut immune dysfunction promoting the translocation of gut-derived bacteria and endotoxins, known as the “leaky gut theory”, have been suggested to induce systemic inflammation in CKD patients [1,25,26]. This theory has also been proposed as an explanation for the systemic inflammation in several other non-intestinal diseases such as cirrhosis, chronic viral hepatitis, non-alcoholic fatty liver disease, obesity, diabetes mellitus, heart failure, HIV infection and diverse autoimmune diseases like rheumatoid arthritis [18,19,27]. The present prospective clinical study shows that stage I–IV CKD and ESKD are associated with decreased intestinal expression of the key TJ molecular components occludin and claudin-1, implying a potential cellular mechanism of gut barrier dysfunction leading to endotoxemia and systemic inflammation. Previous experimental studies with CKD animals have demonstrated the disruption of intestinal TJs [24]. A previous retrospective clinical study also demonstrated reduced expression of TJ proteins in the colonic mucosa of advanced CKD and hemodialysis patients complicated with intradialytic hypotension [28]. However, in this previous study, samples were taken after a bowel operation (colectomy) was performed due to malignancy or acute inflammatory conditions (bowel obstruction, ischemic necrosis, perforation, diverticulitis). Since all of these conditions may affect intestinal TJ expression and function [19], in the present study, these procedures were set as exclusion criteria to reduce potential confounders and attribute the TJ immunohistochemical results to the underlying CKD. According to our results, a specific expression pattern of occludin was observed in the intestinal epithelium in stage I–IV CKD and ESKD patients, with a gradually increasing loss of its expression from the crypt to the tip of the villi. The latter may be explained by the fact that the villus tip is the most vulnerable part to ischemic alterations, which are frequently encountered in CKD patients, and ischemia is an important factor promoting disruption of epithelial TJs [19,29].

Decreased intestinal occludin and claudin-1 expression might be attributed to diverse factors in CKD. Chronic kidney disease leads to uremic toxins’ accumulation (indoxyl sulphate, indole-3 acetic acid, p-cresyl sulphate, trimethylamine/trimethylamine-N-oxide, and phenylacetylglutamine), which might affect TJ structure and function [30,31,32,33]. Additionally, the dietary restrictions, as well as the medications used for the treatment of CKD complications, chronic antibiotic administration due to infections, hemodynamic alterations during hemodialysis and metabolic acidosis, all may lead to gut microbiota alterations [28,31,32,34,35,36]. The normal gut microbiota, through the production of beneficial metabolites (e.g., short chain fatty acids (SCFA), such as butyrate, propionate and acetate), provides energy to enterocytes, while gut dysbiosis is associated with epithelial dysfunction and disruption of enterocyte TJs [19,37].

Disruption of epithelial TJs in the intestine facilitates the paracellular transport of potentially harmful hydrophilic molecules from the intestinal lumen to bloodstream (18). It has been demonstrated that endotoxin, a lipopolysaccharide (LPS) located in the bacterial outer membrane, is increased in all CKD stages as well as in patients with ESKD receiving dialysis [38,39]. This is in line with the findings of the present study, where patients with CKD of various stages had significantly higher endotoxin levels compared to healthy subjects. In addition, according to our findings, serum endotoxin levels in patients with stage I–IV CKD were not different from those of patients with ESKD, which indicates that gut barrier dysfunction is not only a late renal disease complication. This is also supported by the observed similar magnitude of disruption of intestinal TJs between CKD and ESKD patients. The significant inverted correlation of occludin and claudin-1 expression with serum endotoxin concentration points towards a causal relationship between the disruption of enterocyte TJs and systemic endotoxemia in CKD. Previous studies have also demonstrated that increased intestinal permeability directly correlates with endotoxemia [40].

In CKD, endotoxin is an additional stimulus in the development of a systemic inflammatory response [41]. Endotoxin stimulates the production of reactive oxygen species (ROS) from neutrophils and macrophages and induces the production of diverse proinflammatory cytokines such as IL-1, TNF-α and IL-6, through the nuclear factor kappa B (NF κB) signaling pathway [7,42,43,44]. In the present study, IL-6, -8 and -10 were significantly increased in stage I–IV CKD and ESKD patients. Cytokinemia might represent an additional factor implicated in the disruption of intestinal TJs [19]. Specifically, for occludin, previous studies have shown that proinflammatory mediators downregulate its promoter [45]. Although programmed hemodialysis in ESKD has been proposed as a contributing factor of systemic inflammation [46], the present study found non-significant differences in cytokines levels between ESKD and CKD non-dialysis dependent patients. This indicates that factors other than the extracorporeal procedure of dialysis might be responsible for the inflammatory response, such as the disruption of the intestinal barrier and the promotion of systemic endotoxemia, as shown in the present study.

Potential therapeutic strategies to control intestinal hyperpermeability in CKD and ESKD patients include (a) interventions to prevent or restore intestinal dysbiosis, which is associated with TJ disruption, with the use of probiotics, prebiotics and synbiotics [19,47]. Also, there is currently a growing research interest on the potential beneficial role of fecal microbiota transplantation in diverse pathological entities characterized by intestinal dysbiosis, gut hyperpermeability and systemic inflammation [48]. This could be an interesting research field for the future. These strategies also include (b) interventions aiming at preventing or restoring intestinal barrier injury, such as immunonutrition and antioxidants supplementation might also have a positive impact in this direction [49], and (c) interventions to suppress systemic inflammation with the use of anticytokine therapies on a personalized basis according to the observed cytokine profile.

Our study has certain limitations; first, it is a single-center study with a small number of patients. Second, intestinal permeability in patients with CKD or ESKD was only indirectly studied by using endotoxemia as a marker. Third, the potential mechanisms of disruption of enterocytes TJs were not investigated in depth and interrelations of our findings were mostly based on a theoretical basis. However, despite these limitations, our study provides novel insights in the cellular alterations associated with gut barrier dysfunction in CKD patients, highlighting the important role of enterocytes’ TJs.

## 5. Conclusions

In conclusion, the present pilot study demonstrates that occludin and claudin-1 expression in enterocytes is significantly reduced in CKD and ESKD patients. The changes in intestinal TJs may represent an important cellular mechanism for gut barrier dysfunction in CKD, leading to increased gut permeability, endotoxemia and systemic inflammation. Uncovering the molecular basis of increased gut permeability might lead to future pharmacological studies focusing on the modulation of intestinal TJs, which could lead to better control of intestinal hyperpermeability in CKD patients, thus improving clinical outcomes.

## Figures and Tables

**Figure 1 biomedicines-12-00368-f001:**
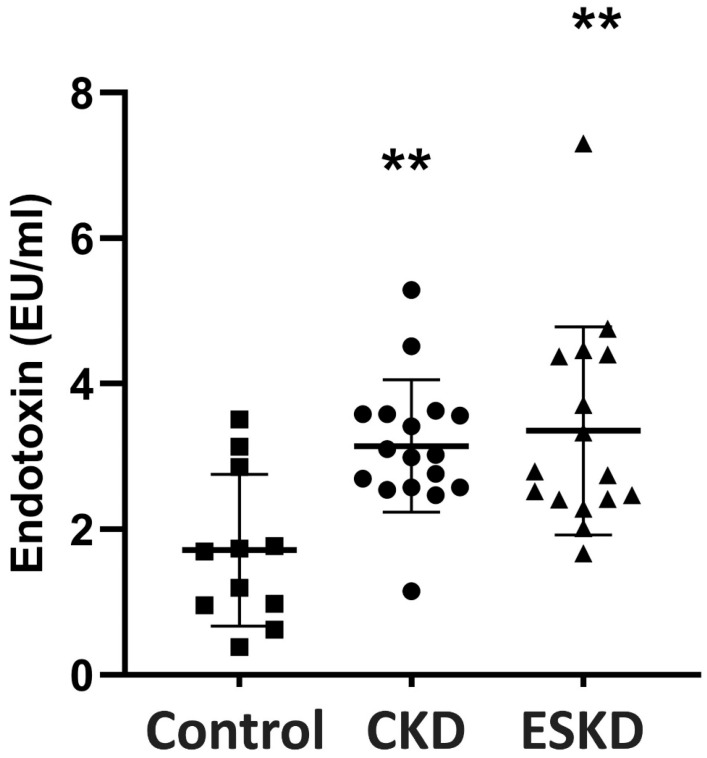
Endotoxin concentrations in the peripheral blood of CKD patients and controls. Statistical analysis was performed with nonparametric analysis of variance (Kruskal–Wallis test) followed by a post hoc Mann–Whitney U test with Bonferroni correction. CKD: chronic kidney disease; ESKD: end stage kidney disease. ** *p* < 0.01 vs. controls.

**Figure 2 biomedicines-12-00368-f002:**
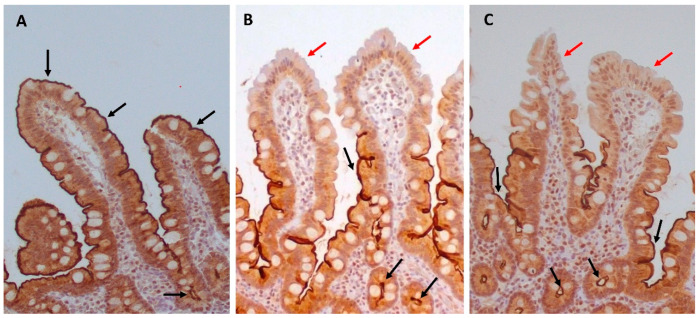
Representative photomicrographs of occludin immunohistochemical expression in the duodenal mucosa: in controls (**A**), almost all of the epithelial cells lining villi and crypts exhibit positive cytoplasmic and apical membranous immunostaining for occludin (black arrows). In patients with CKD (**B**) and ESKD (**C**), there is a pronounced depletion of cytoplasmic immunostaining for occludin and loss of apical membranous expression in numerous epithelial cells, mainly at the upper part of the villi (red arrows), while expression is generally preserved in crypts (black arrows) (all microphotographs ×200).

**Figure 3 biomedicines-12-00368-f003:**
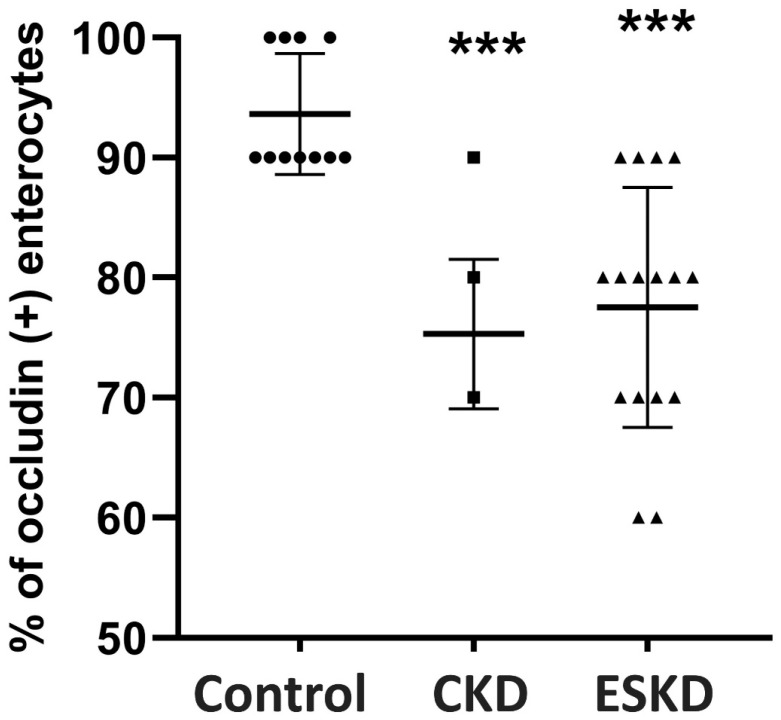
Occludin expression (% of occludin (+) enterocytes) in the intestinal mucosa of controls and CKD patients. Statistical analysis was performed with nonparametric analysis of variance (Kruskal–Wallis test) followed by a post hoc Mann–Whitney U test with Bonferroni correction. CKD: chronic kidney disease: ESKD: end stage kidney disease. *** *p* < 0.001 vs. controls.

**Figure 4 biomedicines-12-00368-f004:**
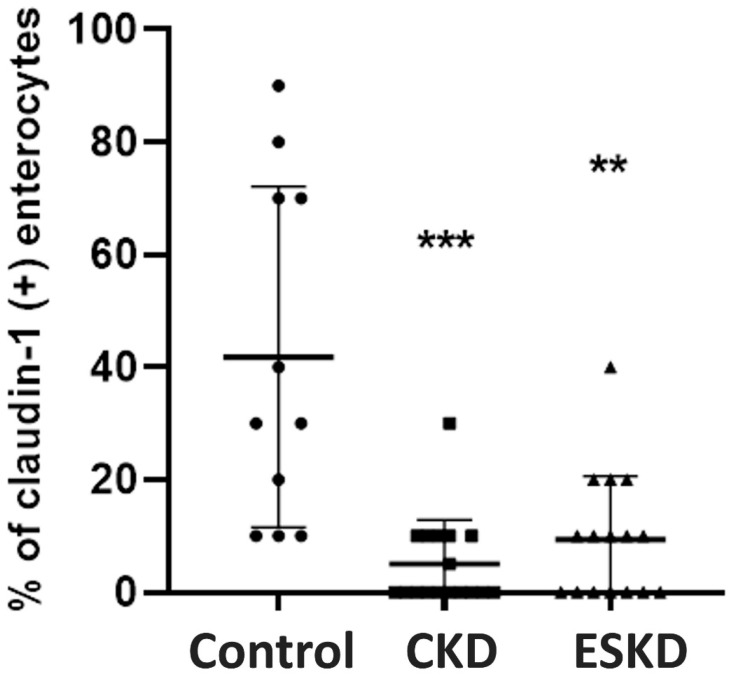
Claudin-1 expression (% of claudin-1 (+) enterocytes) in the intestinal mucosa. Statistical analysis was performed with nonparametric analysis of variance (Kruskal–Wallis test) followed by a post hoc Mann–Whitney U test with Bonferroni correction. CKD: chronic kidney disease; ESKD: end stage kidney disease. ** *p* < 0.01, *** *p* < 0.001 vs. controls.

**Table 1 biomedicines-12-00368-t001:** Characteristics of patients with CKD or ESKD and healthy controls.

Characteristics	Control (n = 11)	Stage I–IV CKD (n = 17)	ESKD (n = 16)
Age (years)	49.7 ± 18.5	58.5 ± 12.5	57.7 ± 13.4
Sex (males/females)	5/7	10/7	11/5
Urea (mL/dL)	30 ± 3.9	63.8 ± 39 *	123.5 ± 39 **
Creatinine (mL/dL)	0.75 ± 0.09	0.75 ± 0.4 *	5.9 ± 2.3 **
eGFR (mL/min/1.73 m^2^)	106.6 ± 19	56.5 ± 23.7 **	<15
Causes of CKD			
Diabetic Nephropathy	N/A	2	5
Hypertensive Nephropathy	N/A	0	4
Glomerulonephritis	N/A	13	2
Interstitial Nephritis	N/A	0	2
Other	N/A	2	1
Unknown	N/A	0	2
CKD Classification			
Stage I (eGFR ≥ 90)	N/A	3	0
Stage II (eGFR = 60–89)	N/A	2	0
Stage III_A_ (eGFR = 45–59)	N/A	7	0
Stage III_B_ (eGFR = 30–44)	N/A	4	0
Stage IV (eGFR = 15–29)	N/A	1	0
Stage V (eGFR < 15)	N/A	0	16
Renal replacement method			
Hemodialysis			10
Peritoneal dialysis			6

Statistical analysis was performed with one-way analysis of variance followed by a post hoc Tukey test for numerical data (age, Urea, Creatinine, eGFR) and with the chi-squared test, with Yates’ correction for the proportional data (sex). * *p* < 0.05, ** *p* < 0.01 vs. Controls. N/A: not applicable.

**Table 2 biomedicines-12-00368-t002:** Cytokine levels in peripheral blood (values are median (IQR)).

Cytokine	Controls (n = 11)	Stage I–IV CKD (n = 17)	ESKD (n = 16)
IL-1β (pg/mL)	0 (0–0.53)	0 (0–0)	0 (0–0)
IL-6 (pg/mL)	0.83 (0.57–1.85)	10.36 (8.35–19.36) **	19.33 (0.58–59.43) **
IL-8 (pg/mL)	3.42 (2.5–6.94)	41 (14.87–102) *	126.3 (33.9–475) ***
IL-10 (pg/mL)	0.28 (0.18–0.37)	2.8 (0.96–4.23) ***	1.85 (0.64–3.1) **
TNF-a (pg/mL)	0.41 (0–1.3)	0 (0–4.7)	0 (0–4)

Statistical analysis was performed with nonparametric analysis of variance (Kruskal–Wallis test) followed by a post hoc Mann–Whitney U test with Bonferroni correction. * *p* < 0.05, ** *p* < 0.01, *** *p* < 0.001 vs. Controls. There were no significant differences between stage I–IV CKD and ESKD groups for all parameters.

**Table 3 biomedicines-12-00368-t003:** Histological features in the intestinal mucosa of stage I–IV CKD or ESKD patients and controls (values are median (IQR), except for villous length that are mean ± SD).

Histopathological Features	Controls (n = 11)	Stage I–IV CKD (n = 17)	ESKD (n = 16)
Apoptotic body count	10 (5–10)	5 (5–10)	7.5 (5–10)
Villous length (mm)	0.42 ± 0.03	0.40 ± 0.13	0.34 ± 0.1
Intraepithelial CD3+ lymphocytes/100 intestinal epithelial cells	15 (15–20)	5 (5–7.5)	10 (5–17.5)

Statistical analysis was performed with nonparametric analysis of variance (Kruskal–Wallis test) for apoptotic body count and intraepithelial CD3+ lymphocytes or with one-way analysis of variance for villous length. There were no significant differences between the study groups.

**Table 4 biomedicines-12-00368-t004:** Gradient of occludin expression (% of occludin (+) enterocytes) along the crypt–villous axis in patient groups (values are mean ± SD).

Occludin Expression	Controls (n = 11)	CKD Stage I–IV (n = 17)	ESKD (n = 16)
**Part of the villi**			
Τip	91.8 ± 7.5	44.7 ± 25 **	53.7 ± 29 **
Μiddle	97.3 ± 4.7	84.7 ± 6.2 **	86.3 ± 7.2 *
Crypt	99 ± 3	97 ± 5.9	93.1 ± 11.3
**Total expression**	93.6 ± 5	75.3 ± 6.2 **	77.5 ± 10 **

Statistical analysis was performed with nonparametric analysis of variance (Kruskal–Wallis test) followed by a post hoc Mann–Whitney U test with Bonferroni correction. * *p* < 0.05, ** *p* < 0.01 vs. Controls.

## Data Availability

Data of this paper are available at Editor’s request.

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
