# Peer review of "Altered Expression of Intestinal Tight Junctions in Patients with Chronic Kidney Disease: A Pathogenetic Mechanism of Intestinal Hyperpermeability"

_biomedicines, 2024, doi:10.3390/biomedicines12020368_

Round 1
Reviewer 1 Report
Comments and Suggestions for Authors
The article is dedicated to an urgent topic and the results obtained by the authors of the article may be useful to the readers of the journal Biomedicines. In the meantime, I have a few comments.
1. Disruption of the intestinal barrier can be both a cause and a consequence of systemic inflammation. The authors may need to consider this pattern in the "Abstract" and "Introduction" sections.
2. Diabetes mellitus, hypertension, primary and secondary glomerulonephritis (caused by systemic lupus erythematosus, systemic vasculitis, myelo and lymphoproliferative disorders and others), polycystic kidney disease, obstructive uropathy, vesicoureteral reflux, renal amyloidosis and drug nephropathy are the most common causes of CKD. Many of these diseases may themselves be the cause of chronic systemic inflammation, independent of the presence of CKD [Gusev E, Solomatina L, Zhuravleva Y, Sarapultsev A. The Pathogenesis of End-Stage Renal Disease from the Standpoint of the Theory of General Pathological Processes of Inflammation. Int J Mol Sci. 2021;22(21):11453]. In addition, the use of programmed haemodialysis in ESKD is an independent cause of systemic inflammation. Therefore, in the "Materials and methods" section, the authors should present the nosology of both CKD groups and, in the ESKD group, the percentage distribution of peritoneal and programmed dialysis.
3. In Table 2, it is necessary to indicate the reliability of the differences not only between control and CKD, but also between the two groups of CKD, and to discuss the significance of these differences in the final sections of the article. In Table 2-4, the stages of CKD (I-IV) must be indicated.
4. In the Discussion section, the authors state the following: "To our knowledge, this is the first clinical study demonstrating that stage I-IV CKD and ESKD are associated with decreased intestinal expression of the key TJ molecular components occludin and claudin-1, suggesting a potential cellular mechanism of intestinal barrier dysfunction leading to endotoxemia and systemic inflammation". This statement is surprising. Probably the authors should read their own reference again and more carefully - [34] in the "References" section.
Author Response
From:
Stelios F. Assimakopoulos, MD, PhD, Associate Professor, Department of Internal Medicine, Division of Infectious Diseases, University of Patras Medical School, Patras 26504, Greece.
sassim@upatras.gr
To: BIOMEDICINES
Patras, January 25, 2024
Dear Editor,
First, we would like to wish you a Happy New Year, full of health, happiness and successes!
Thank you for reviewing our manuscript entitled “Altered Expression of Intestinal Tight Junctions in Patients with Chronic KIDNEY disease: A Pathogenetic Mechanism of Intestinal Hyperpermeability”. We appreciate the time and efforts that you and the reviewers dedicated to providing valuable comments on our article.
We have now completed the revisions required by the reviewers. All changes are highlighted in the revised manuscript. A point-by point answer to the comments made by the reviewers is provided below.
The revised manuscript has been seen and approved by all authors,
We hope that our revised manuscript will meet your requirements for publication in your respectable journal.,
Sincerely,
Stelios F. Assimakopoulos, MD, PhD
Associate Professor Internal Medicine/Infectious Diseases
Point by point reply to comments of Reviewer 1
Comments to author
The article is dedicated to an urgent topic and the results obtained by the authors of the article may be useful to the readers of the journal Biomedicines. In the meantime, I have a few comments.
Dear reviewer, thank you for your positive evaluation on our work.
Comment 1.
Disruption of the intestinal barrier can be both a cause and a consequence of systemic inflammation. The authors may need to consider this pattern in the "Abstract" and "Introduction" sections.
Reply: Thank you for this comment. This bidirectional relation between inflammation and gut barrier dysfunction has been indicated clearer now in the abstract (lines 24-26) and the Introduction section (lines 49-52).
Comment 2.
Diabetes mellitus, hypertension, primary and secondary glomerulonephritis (caused by systemic lupus erythematosus, systemic vasculitis, myelo and lymphoproliferative disorders and others), polycystic kidney disease, obstructive uropathy, vesicoureteral reflux, renal amyloidosis and drug nephropathy are the most common causes of CKD. Many of these diseases may themselves be the cause of chronic systemic inflammation, independent of the presence of CKD [Gusev E, Solomatina L, Zhuravleva Y, Sarapultsev A. The Pathogenesis of End-Stage Renal Disease from the Standpoint of the Theory of General Pathological Processes of Inflammation. Int J Mol Sci. 2021;22(21):11453]. In addition, the use of programmed haemodialysis in ESKD is an independent cause of systemic inflammation. Therefore, in the "Materials and methods" section, the authors should present the nosology of both CKD groups and, in the ESKD group, the percentage distribution of peritoneal and programmed dialysis.
Reply: We would like to thank the reviewer for this remark. We totally agree that the primary disease and cause of CKD like certain types of glomerular diseases (especially Lupus or vasculitis) and diabetes are causes of increased systemic inflammation and interpret to some point the increase of serum cytokine levels showed in this group of patients. However, this is a factor that could not be eliminated in a clinical study. According to your right suggestion the underlying pathology of both CKD and ESKD groups has been added now in the revised Table 1 (line 178). We also agree with the reviewer that extracorporeal dialysis is a contributing factor of systemic inflammation (reference added: Gusev E. et al. Int J Mol Sci. 2021;22(21):11453). However, in our study, the finding of non-significant differences in cytokine levels between ESKD and CKD non-dialysis dependent patients (see Results section and Table 2), indicates that factors other than the extracorporeal procedure of dialysis might be responsible for the inflammatory response, such as the disruption of the intestinal barrier and the promotion of systemic endotoxemia. This has been commented now in the Discussion section (lines 326-332).
Comment 3.
In Table 2, it is necessary to indicate the reliability of the differences not only between control and CKD, but also between the two groups of CKD, and to discuss the significance of these differences in the final sections of the article. In Table 2-4, the stages of CKD (I-IV) must be indicated.
Reply: There were no significant differences between CKD stage I-IV and ESKD groups for all cytokines and this has been written clearly now in the Table 2 footnote (line 200-201) and in the corresponding section of the results (lines 205-206). A relative comment has been added in the Discussion section (lines 326-332). Also, the stages of CKD (I-IV) were indicated now in tables 2-4.
Comment 4.
In the Discussion section, the authors state the following: "To our knowledge, this is the first clinical study demonstrating that stage I-IV CKD and ESKD are associated with decreased intestinal expression of the key TJ molecular components occludin and claudin-1, suggesting a potential cellular mechanism of intestinal barrier dysfunction leading to endotoxemia and systemic inflammation". This statement is surprising. Probably the authors should read their own reference again and more carefully - [34] in the "References" section.
Reply: Thank you for this notion, which was an oversight from our part. The relative part of the discussion was corrected, and the findings of this related previous work were described in the revised text (lines 277-280). However, in this previous study samples were taken after bowel operation (colectomy) performed due to malignancy or acute inflammatory conditions (bowel obstruction, ischemic necrosis, perforation, diverticulitis). Since all these conditions may affect intestinal TJ expression and function (Assimakopoulos SF, et al. World J Gastrointest Pathophysiol 2011;2(6):123-137), in the present study were set as exclusion criteria to reduce potential confounders and attribute the TJ immunohistochemical results to the underlying chronic kidney disease. This important difference between the two studies was also commented in the Discussion (lines 280-285).
Reviewer 2 Report
Comments and Suggestions for Authors
This is an interesting study about intestinal hyperpermeability in CKD.
It needs some brushing up regarding the statistical calculations and the manner these are reflected in tables in figures.
Table 2 and 3: the significance of the numbers in columns 2, 3, and 4 should be explained in the table caption (as they are in the caption of Table 4) - may be they are median and interquartile range or mean±SD
Figure 5 is uninformative and drawing a regression line is inappropriate as visual inspection does not suggest a linear relationship between x and y variables (especially in the lower panel); maybe a hyperbolic one. Anyway, my suggestion is to drop this figure or at least the regression line.
For each table and figure in which p values are mentioned the statistical tests employed for calculating the p values should be clearly indicated.
It is not clear for which calculations Chi square test was used - I cannot identify a table or figure the data of which may be subjected to Chi square test.
Author Response
From:
Stelios F. Assimakopoulos, MD, PhD, Associate Professor, Department of Internal Medicine, Division of Infectious Diseases, University of Patras Medical School, Patras 26504, Greece.
sassim@upatras.gr
To: BIOMEDICINES
Patras, January 25, 2024
Dear Editor,
First, we would like to wish you a Happy New Year, full of health, happiness and successes!
Thank you for reviewing our manuscript entitled “Altered Expression of Intestinal Tight Junctions in Patients with Chronic KIDNEY disease: A Pathogenetic Mechanism of Intestinal Hyperpermeability”. We appreciate the time and efforts that you and the reviewers dedicated to providing valuable comments on our article.
We have now completed the revisions required by the reviewers. All changes are highlighted in the revised manuscript. A point-by point answer to the comments made by the reviewers is provided below.
The revised manuscript has been seen and approved by all authors,
We hope that our revised manuscript will meet your requirements for publication in your respectable journal.,
Sincerely,
Stelios F. Assimakopoulos, MD, PhD
Associate Professor Internal Medicine/Infectious Diseases
Point by point reply to comments of Reviewer 2.
Comments to author
This is an interesting study about intestinal hyperpermeability in CKD.
It needs some brushing up regarding the statistical calculations and the manner these are reflected in tables in figures.
Dear reviewer, thank you for your positive evaluation on our work.
Comment 1.
Table 2 and 3: the significance of the numbers in columns 2, 3, and 4 should be explained in the table caption (as they are in the caption of Table 4) - may be they are median and interquartile range or mean±SD.
Reply: Tables 2 and 3 were revised according to your suggestion and now it is clearly stated in the caption what the values represent (lines 197 and 213-214).
Comment 2.
Figure 5 is uninformative and drawing a regression line is inappropriate as visual inspection does not suggest a linear relationship between x and y variables (especially in the lower panel); maybe a hyperbolic one. Anyway, my suggestion is to drop this figure or at least the regression line.
Reply: Thank you for your comment. To avoid any inaccurate representation of our correlation results we followed your suggestion and omitted the figure 5.
Comment 3.
For each table and figure in which p values are mentioned the statistical tests employed for calculating the p values should be clearly indicated.
Reply: The statistical tests employed in each table and figure have been clearly stated now in all tables’ footnotes and in all figure captions.
Comment 4.
It is not clear for which calculations Chi square test was used - I cannot identify a table or figure the data of which may be subjected to Chi square test.
Reply: Chi square test was only used in table 1 to compare the proportional data of gender distribution. This has been stated now in the Table 1 footnote (line 179-181).
Reviewer 3 Report
Comments and Suggestions for Authors
The paper titled: Altered expression of intestinal tight junctions in patients with chronic kidney disease: a pathogenetic mechanism of intestinal hyperpermeability (biomedicines-2843105) describes the associations between CKD and parameters of the intestinal barrier – such as tight junction proteins such as occludin and claudin-1 in the intestinal epithelium and serum cytokines IL-1β, IL-6, IL-8, IL-10, and TNF-α. The Authors found that intestinal occludin and claudin-1 were significantly decreased and correlated with systemic endotoxemia. The Authors also found the specific expression pattern of occludin in the villi in CKD patients.
The paper is very interesting and describes; however, a few issues should be explained.
1. Why the patients had endoscopy? There is information that only patients without H.pylori were included in the study, but information about the reason for endoscopy is included. Such information should be also mentioned in Table 1 – patient characteristics.2. There are four groups of integral transmembrane proteins. Why did you decide to measure the occludin and claudin-1 expressions?
3. The “leaky gut theory” (line 271) is very interesting. What other non-renal and non-intestinal diseases are characterized by a “leaky gut”? Such an explanation can show the problem and increase the interest of other scientists and clinicians who treat other inflammatory diseases.
4. What is the difference between interstitial nephritis (2 patients with ESRD) and “leaky gut”?
5. In conclusion, the Authors wrote that the modulation of intestinal TJs is important in preventing and decreasing systemic inflammation. How can the treatment change the modulation of intestinal TJs? Can you suggest addiction treatment strategies, except the standard treatment of CKD, particularly in ESRD?
If the authors consider the above comments, the work is valuable and can be published.
Author Response
From:
Stelios F. Assimakopoulos, MD, PhD, Associate Professor, Department of Internal Medicine, Division of Infectious Diseases, University of Patras Medical School, Patras 26504, Greece.
sassim@upatras.gr
To: BIOMEDICINES
Patras, January 30, 2024
Dear Editor,
First, we would like to wish you a Happy New Year, full of health, happiness and successes!
Thank you for reviewing our manuscript entitled “Altered Expression of Intestinal Tight Junctions in Patients with Chronic KIDNEY disease: A Pathogenetic Mechanism of Intestinal Hyperpermeability”. We appreciate the time and efforts that you and the reviewers dedicated to providing valuable comments on our article.
We have now completed the revisions required by the 3rd reviewer. All changes are highlighted in the revised manuscript. A point-by point answer to the comments made by the reviewer is provided below.
The revised manuscript has been seen and approved by all authors,
We hope that our revised manuscript will meet your requirements for publication in your respectable journal.
Sincerely,
Stelios F. Assimakopoulos, MD, PhD
Associate Professor Internal Medicine/Infectious Diseases
Point by point reply to comments of Reviewer 3.
Comments to author
The paper titled: Altered expression of intestinal tight junctions in patients with chronic kidney disease: a pathogenetic mechanism of intestinal hyperpermeability (biomedicines-2843105) describes the associations between CKD and parameters of the intestinal barrier – such as tight junction proteins such as occludin and claudin-1 in the intestinal epithelium and serum cytokines IL-1β, IL-6, IL-8, IL-10, and TNF-α. The Authors found that intestinal occludin and claudin-1 were significantly decreased and correlated with systemic endotoxemia. The Authors also found the specific expression pattern of occludin in the villi in CKD patients. The paper is very interesting and describes; however, a few issues should be explained.
Dear reviewer, thank you for your positive evaluation on our work.
Comment 1.
Why the patients had endoscopy? There is information that only patients without H.pylori were included in the study, but information about the reason for endoscopy is included. Such information should be also mentioned in Table 1 – patient characteristics
Reply: Thank you for this comment, which will help us describe clearer the relative Method section. All subjects enrolled in the study underwent an upper gastrointestinal tract endoscopy due to symptoms of dyspepsia, after consultation of a gastroenterologist, without any pathological findings. This information has been added now in the “Study design” paragraph (lines 91-95). Since the reason of endoscopy was common for all study subjects, according to the protocol, there is no need to revise table 1 with information about potential different etiologies for endoscopy.
Comment 2.
There are four groups of integral transmembrane proteins. Why did you decide to measure the occludin and claudin-1 expressions?
Reply: Thank you for this comment which will help us better clarify the rationale of our study. Selection of occludin and claudin-1 TJs proteins to be studied in patients with CKD was based on previous in vitro and animal studies demonstrating that uremic conditions and CKD impair gut barrier integrity through degradation of the transcellular occludin and claudin-1 expression. This information has been updated now in the revised text (Introduction section, lines 63-66).
Comment 3.
The “leaky gut theory” (line 271) is very interesting. What other non-renal and non-intestinal diseases are characterized by a “leaky gut”? Such an explanation can show the problem and increase the interest of other scientists and clinicians who treat other inflammatory diseases.
Reply:
The leaky gut theory has been proposed as an explanation of systemic inflammation in several other non-intestinal diseases such as cirrhosis, chronic viral hepatitis, non-alcoholic fatty liver disease, obesity, diabetes mellitus, heart failure, HIV infection and diverse autoimmune diseases like rheumatoid arthritis. This information has been added now in the first paragraph of the Discussion section (lines 272-275).
Comment 4.
What is the difference between interstitial nephritis (2 patients with ESRD) and “leaky gut”?
Reply: We are sorry that we cannot precisely understand this comment. Differences in the leaky gut parameters were not compared between ESKD subtypes, owing to low number of patients.
Comment 5.
In conclusion, the Authors wrote that the modulation of intestinal TJs is important in preventing and decreasing systemic inflammation. How can the treatment change the modulation of intestinal TJs? Can you suggest addiction treatment strategies, except the standard treatment of CKD, particularly in ESRD?
Reply: Dear reviewer, thank you for this comment. Potential therapeutic strategies to control intestinal hyperpermeability in CKD and ESKD patients include: (a) Interventions to prevent or restore intestinal dysbiosis, which is associated with TJs disruption, with the use of probiotics, prebiotics and synbiotics. Also, there is currently a growing research interest on the potential beneficial role of fecal microbiota transplantation in diverse pathological entities characterized by intestinal dysbiosis, gut hyperpermeability and systemic inflammation. This could be an interesting research field for the future. (b) Interventions aiming at preventing or restoring intestinal barrier injury such as immunonutrition and antioxidants supplementation might also have a positive impact in this direction, and (c) Intervention to suppress systemic inflammation with the use of anticytokine therapies on a personalized basis according to the observed cytokine profile. This information has been added now in a new paragraph in the Discussion section (lines 336-346).